# The Effect of Sows’ and Piglets’ Behaviour on Piglet Crushing Patterns in Two Different Farrowing Pen Systems

**DOI:** 10.3390/ani9080538

**Published:** 2019-08-07

**Authors:** Thies Nicolaisen, Eyke Lühken, Nina Volkmann, Karl Rohn, Nicole Kemper, Michaela Fels

**Affiliations:** 1Institute for Animal Hygiene, Animal Welfare and Farm Animal Behaviour, University of Veterinary Medicine Hannover, Foundation, Bischofsholer Damm 15, D-30173 Hannover, Germany; 2Institute for Biometry, Epidemiology and Information Processing, University of Veterinary Medicine Hannover, Foundation, Bünteweg 2, D-30559 Hannover, Germany

**Keywords:** farrowing, loose housing, preweaning mortality, piglet crushing, posture change

## Abstract

**Simple Summary:**

Farrowing crates—narrow cages where sows are kept during lactation—impede the sow in the expression of natural behaviours and, therefore, cause animal welfare concerns. However, piglet losses due to crushing by the sow are effectively reduced by farrowing crates. Hence, there is an urgent need to find a practical compromise between sows’ and piglets’ welfare. The aim of this study was to test two farrowing pens without fixation of the sow in comparison to pens with a farrowing crate. Piglet mortality, piglet crushing and sow and piglet behaviour in the first 72 h after birth were analysed. Piglet mortality was higher due to increased levels of piglet crushing in the free-farrowing pens. However, the majority of crushing occurred in the first three days after birth. The recorded active and resting behaviour of sows in the first 72 h after birth hardly highlighted differences between the three systems, i.e., sows in free-farrowing pens hardly used the offered possibilities for activity in the first 72 h after birth. In conclusion, our results suggest that a temporary fixation for a few days after birth could be sufficient to significantly reduce piglet crushing and could represent a practical solution for future farrowing systems.

**Abstract:**

Pens with farrowing crate (FC) and two differently designed free-farrowing pens (LH-pens: 7.3 m², plastic flooring; GH-pens: 5 m², cast-iron and concrete flooring) were compared regarding piglet losses and postpartum sow behaviour (all treatments) and reasons for piglet crushing and postpartum litter behaviour (LH and GH). One-hundred-and-three crushing events were analysed in eight batches concerning sows’ posture changes that crushed piglets and age of crushed piglets. Posture change frequency, amounts of single posture changes and total time spent in different body postures were evaluated for 41 sows (14 FC-sows, 13 LH-sows and 14 GH-sows) in six batches. Litter behaviour (location, active/inactive scoring, resting behaviour next to the sow) was analysed during sows’ posture changes and piglet crushing. Piglet mortality was higher in LH (25.6%) and GH (19.9%) compared to FC (12.3%) due to higher levels of piglet crushing. Most crushing occurred during the three days postpartum in LH (92.7%) and GH (83.9%). However, crushing patterns differed between LH (rolling: 68.2%; sit-to-lie: 18.2%; stand-to-lie: 11.4%) and GH (rolling: 38.2%; sit-to-lie: 30.9%; stand-to-lie: 16.4%) and varying piglet behaviour may be the cause for this. The postpartum period was characterized by inactivity of the sow and behavioural differences were rarely seen between systems.

## 1. Introduction

Crates for sows are banned for most of the gestation period in the European Union [1], with mandatory group housing in the period from four weeks after insemination to one week before farrowing. However, pens with farrowing crates are still permitted and, moreover, are a widespread housing system for lactating sows in many countries. The farrowing crate was developed to reduce piglet mortality due to crushing, to improve the farrowing environment with regard to piglet survival and to simplify human interventions at farrowing [2]. It offers economic advantages regarding space requirements, time management and occupational safety. However, keeping sows in farrowing crates raises animal welfare concerns because the farrowing crate impedes the sow in the expression of natural behaviours [3]. Furthermore, sows in farrowing crates are more susceptible to stereotypical behaviour (e.g., vacuum chewing or bar biting [4,5]). In addition to this, society is becoming more and more sensitized to the handling of farm animals and claims for higher animal welfare standards in livestock husbandry. Even if the following studies cannot predict the actual behaviour of meat consumers, former researchsuggests that consumers are willing to spend more money on meat that was produced under improved animal welfare conditions [6,7].

A multitude of free-farrowing pens have been designed and tested [8], but increased pre-weaning piglet mortality remains the major problem in systems with unrestricted sows [9,10]. Modern pig breeding has transformed a relatively lightweight wild sow with a small litter-size into an over three hundred kilogram heavy and hyperprolific domestic sow with physically undeveloped and vulnerable offspring. These piglets are highly susceptible to hypothermia, starvation and, subsequently, to crushing, which is the main cause of pre-weaning piglet mortality in intensive pig husbandry [2]. Crushing is the result of a complex interaction between sow and piglet behaviour. It is known that individual behavioural differences of sows and piglets influence the risk for piglet crushing; for example, the manner of lying down behaviour and reaction of the sow to piglets’ distress calls [11], or the rooting behaviour of the sow prior to lying down events [12]. Furthermore, Roehe and Kalm [13] discovered that piglets with low birth weights are at higher risk of pre-weaning mortality. It was observed that lightweight piglets behave more carelessly, which could increase the probability of being crushed [14]. Another important factor that can influence the extent of piglet crushing is the design of the farrowing pen. Several attempts were made to protect piglets in free-farrowing pens, e.g., sloped flooring, protection rails or a separated creep area [8]. However, there is no general solution for a reliable prevention of piglet crushing in free-farrowing pens. The most dangerous period for piglets with regard to crushing is the four-day-period after parturition [15]. Additionally, it is known that piglets in loose-housing pens are mainly crushed either when moving from a standing position to lying or by the rolling behaviour of the sow, i.e.,a transition from sternal to lateral recumbency or vice versa [16]. Both behaviour patterns can be effectively restricted by a farrowing crate and are influenced by the housing system [17].

On the one hand, banning farrowing crates can be expected to be beneficial for the sows’ welfare. However, on the other hand, higher levels of crushed piglets in free-farrowing pens would restrict piglets’ welfare and lead to considerable economic losses for the farmers. Up to now, only a few studies have shown that using a loose-housing system for lactating sows is possible without causing higher pre-weaning mortality [18,19,20]. For this reason, and because maternal behaviour can only be influenced by breeding to a certain degree [21], it is necessary to develop new solutions for free-farrowing pens and to improve their existing designs.

The present study tested two differently designed (size, flooring and arrangement of pen equipment) prototypes of free-farrowing pens with regard to piglet crushing. Therefore, it was recorded in which ways sows crushed their piglets in the two different single loose-housing pens. Frequencies and types of sows’ body posture changes during 72 h postpartum were evaluated and compared between sows in farrowing crates and sows in the two free-farrowing pens. Furthermore, the influence of sows’ body posture changes during the postpartum period and piglets’ behaviour during these’ posture changes on the detected crushing patterns in the two free-farrowing pens was examined.

## 2. Materials and Methods

### 2.1. Animals and Housing Conditions

The experiments were conducted on the research farm for pig breeding and pig husbandry of the Chamber of Agriculture for Lower Saxony in Wehnen, Germany, between July 2016 and August 2017.

All of the animals were housed in accordance with EU (European Directive 2008/120/EC) and national law (Tierschutzgesetz, Tierschutz-Nutztierhaltungs-Verordnung). In compliance with European Directive 2010/63/EC Article 1 5. (f), the present study did not imply any invasive procedure or treatment to the animals. The study was reviewed and approved by the Animal Welfare Officer of the University of Veterinary Medicine Hannover, Foundation, Germany.

The sows participating in the experiment were a crossbreed between Landrace and Large White (genetics: BHZP, db.Viktoria). Before being moved to the different farrowing systems, sows were kept in groups of four to five in gestation pens. All multiparous sows had farrowed in pens with farrowing crates in their previous gestations.

Three different farrowing systems were tested: pens with farrowing crates (FC), single loose housing pens (LH) and a group-housing system for six lactating sows (GH) that contained six lockable pens and a central common area. Two rooms existed for each farrowing system (FC: eight pens per room; LH and GH: six pens per room) and all rooms were located in the same building. All pen elements of the three farrowing systems were provided by Big Dutchman International GmbH, Vechta, Germany.

In total, the FC-pens (Figure 1) were 260 cm long and 200 cm wide, the dimensions of the farrowing crate were 190 cm × 80 cm. An open creep area, heated by an infrared light, existed (160 cm × 50 cm). A fully slatted plastic floor was installed in the pens and tiles were recessed in the sow’s lying area to ensure thermal conduction.

The LH-pens (Figure 2) were 270 cm long and 270 cm wide (7.3 m²). Each pen was divided by a swing gate that limited the space that was actually accessible for the sow. The closed creep area (100 cm × 80 cm) was located in the area of the pen which was not accessible for the sow and it was heated by an infrared radiation element (CE-REX^TM^ IRX-300, Rexlan Europe, Sorö, Denmark). The creep area had a large opening on the front side and a small opening on its lateral side. The floor in the LH-pens was similar to the FC-pens (plastic flooring).

The group housing system for six lactating sows (Figure 3) consisted of two opposite rows of three pens each and a central common area. The pens were 202 cm long and 245 cm wide (5 m²)and a lockable gate granted access to the common area. An approximately 20 cm high roll at the bottom of the gate prevented younger piglets from being able to pass through it.However, a small and closable opening on the front side of the pen allowed piglets to enter the common area. The flooring was different between GH-pens and LH-pens. It was bisected in a fully slatted cast-iron floor (125 cm × 202 cm in the front) and a fully slatted concrete floor (120 cm × 202 cm in the rear area of the pen). The creep area was similar to LH-pens. In contrast to LH, the entire GH-pen was accessible to the sow.

In total, the common area was 610 cm long and 235 cm wide, but one part of it was only accessible for the piglets (100 cm × 235 cm) for pelleted creep feed provision. The floor of the common area was a fully slatted concrete floor. Nipple drinkers for sows and piglets were installed in the pens and in the common area, whereas sows were fed only in their own pen.

Manipulable materials for sows and piglets were offered before birth and throughout lactation: One jute sack was offered to each sow before birth. After birth, two cotton ropes of different diameters were offered to sows and piglets during the entire lactation period and were continuously renewed. In addition, straw was provided in straw racks in the LH- and GH-pens.

LH- and GH-sows were unrestrained both before and after birth. Anti-crushing bars were installed in the LH- and GH-pens and in the common area of the group-housing system.

Sows were fed with a standard diet for lactating sows (14.7 MJ ME/kg, 16.0% crude protein). FC- and LH-sows were fed twice a day (at 08:00 h and at 17:00 h), whereas GH-sows were fed on demand by an electronic feeder. Therefore, the day was divided into different feeding periods (between 08:00 h and 20:00 h) during which GH-sows (tagged with an electronic ear tag) could receive a certain percentage of their daily feed on demand. On the first day postpartum, the sows received 3.0 kg of feed. This amount increased to 7.0 kg on day 9 postpartum and, subsequently, to 8.5 kg on day 15 postpartum. This was the maximum amount of feed for FC- and LH-sows, whereas GH-sows were able to receive up to 9.0 kg from day 17 postpartum onwards.

### 2.2. Farrowing, Cross-Fostering and Piglet Treatment

The sows were moved to the different farrowing systems one week before the expected farrowing date. The lactation period lasted four weeks. While no specific treatment existed for FC- and LH-sows, GH-sows were kept in the single pens of the group housing system for the first 24 h after moving in. After this period, the pens were opened for the following 48 h to enable socialization among the sows in the common area. Subsequently, the sows were penned in again to ensure that farrowing occurred in the pens (separation period). The GH was opened for sows and piglets when the last-born litter was five days old, and it stayed open for the remaining lactation period. Sows and piglets could move freely between all pens and the common area.

During the experiment, a daily work routine was maintained to test the farrowing systems under practical conditions. Daily work procedures were conducted as fast as possible and in the same way in all three farrowing systems to guarantee comparability between systems.

If cross-fostering was necessary, it was performed within 48 h after birth and only within the same farrowing system. It took place when the litter control was conducted after birth. Litter control occurred on the first day postpartum and included weighing the piglets, ear tagging and teeth clipping. Tail docking was done under veterinary advisement on day 4 postpartum, as well as castration of male piglets.

### 2.3. Video Recording

In total, 26 cameras (Everfocus ez.HD, Everfocus, New Taipei City, Taiwan) and two video recorders (EverFocus ECOR FHD 16 × 1, Everfocus, New Taipei City, Taiwan) were installed to record sows’ and piglets’behaviour continuously. In both GH-compartments, all six pens were monitored with one camera each and the common area was recorded with two cameras. Three of six LH-pens (one camera per pen) and four of eight FC-pens (one camera for two pens) were video recorded in each compartment.

### 2.4. Piglet Losses

Numbers of piglets born alive and weaned piglets were evaluated for nine batches in the three systems. Moreover, total piglet losses and the proportion of crushed piglets were assessed for 2338 piglets born alive (FC: n = 791; LH: n = 696; GH: n = 851) in nine batches. Therefore, the trained staff examined each dead piglet to identify piglets that had died due to crushing (signs of traumatic injuries, e.g., bruises or visible impressions of the slatted floor on the piglet’s body).

### 2.5. Piglet Crushing

Piglet crushing was analysed for LH- and GH-sows in eight batches. In total, 70 sows (24 LH-sows, mean parity and standard error of the mean (SEM): 2.96 (±0.47); 46 GH-sows, mean parity and SEM: 2.62 (±0.30)) were video recorded continuously. The trained personnel examined every dead piglet in terms of signs for crushing. In total, 149 possible crushing events were detected and subsequently searched for in the video recordings. One hundred and three piglet crushing events (LH: n = 44 of 14 sows, GH: n= 59 of 28 sows) were confirmed and analysed, whereas 46 events were not evaluable for different reasons (e.g., technical problems or not clearly identifiable events).

The 103 confirmed crushing events were analysed regarding the main types of sows’ posture changes resulting in crushed piglets (stand-to-lie, sit-to-lie, rolling (body posture changes while lying); others) and the age of the crushed piglets in days. Therefore, the time of birth of the first piglet was determined by video observation and set as life hour zero of the litter. Additionally, sub-categories of sows’ body posture changes that crushed piglets (Appendix A) and litter behaviour during the crushing event (Appendix A) were evaluated. Since the focus was laid on a comparison of the pens, only the crushing events that occurred during the separation period in the GH-pens were regarded for further analysis.

### 2.6. Sow and Piglet Behaviour 72 H Postpartum

Quantity and type of body posture changes in the first 72 h postpartum were analysed for 41 sows of six batches in three different farrowing systems (14 FC-sows, 13 LH-sows, 14 GH-sows; mean parities and SEM: FC= 3.0 (±0.61); LH= 3.0 (±0.71); 14 GH= 2.64 (±0.55)). The initial number of observed sows per batch was equal in all three farrowing systems. However, one LH-sow was excluded from the experiment due to obvious lameness. Nine different types of body posture changes were distinguished (Appendix A). Additionally, the exact time of each body posture change was recorded to get an indication of the total time that sows spent in different body postures. The starting point of the 72 h observation period was the birth of the first piglet.

Piglet behaviour in the first three days postpartum was recorded in a sub-sampling that included the litters of eight GH- and nine LH-sows (mean parities and SEM: LH= 2.2 (±0.74); GH= 2.6 (±0.89)) in four batches. The recorded parameters of piglet behaviour were the same as in the category piglet crushing (Appendix A). Piglet behaviour was recorded from the seventh hour of life onwards to ensure that birth was already or almost finished. Within the first day postpartum, the sow was weighed once and the litter control took place. Body posture changes that occurred due to a disturbance in the pen (e.g., personnel cleaning the pen, removal of a dead piglet) were recorded, but not used for further evaluation.

### 2.7. Statistical Analysis

Statistical analysis was performed using commercial software SAS 9.4 and the Enterprise Guide Client 7.15 (SAS Institute Inc., Cary, NC, USA). The level of statistical significance was set at *p* < 0.05.

First, data were tested for normal distribution using histograms and a Shapiro–Wilk test to determine a suitable statistical model for the evaluation. Afterwards, the dependent variables concerning piglet losses and two of the three sow behaviours 72 h postpartum (frequencies of single body posture changes and total time spent in different body postures) were analysed with a generalized linear mixed model for mixture distribution (PROC GLIMMIX).The farrowing system was included as fixed factor and the sows´ parity was included as additional fixed factor for the analysis of sow behaviour. Therefore, sows were divided into three parity classes (first parity; second parity; third or higher parity). The random effect consisted of the individual sow. Multiple comparisons of least squares means (LSMEANS) were calculated using Tukey–Kramer tests.

The body posture change frequency of the sows, for which time comparisons within the farrowing system were conducted, was analysed using a three-way mixed model analysis of variance with farrowing system and parityas fixed effects and examination day with subject-ID sow as fixed effect with repeated measurements, taking into account all interaction terms between effects. Due to significant three-way interaction between the three effects, the influence ofparity, examination day and the interaction of parityand examination day were calculated for each farrowing system separately using a two-way mixed model analysis of variance. Therefore, paritywas included as a fixed effect and the examination day was included as a fixed effect with repeated measurements.

With regard to piglet behaviour, a logistic regression model with random and repeated measurements was carried out for the three categorical, dichotomous effects piglet location, active/inactive scoring and resting behaviour next to the sow before rolling behaviour as dependent variables, farrowing system as independent variable and examination day as fixed effects (explanation variables) with repeated measurements, taking into account also the interaction term between effects. Measurements on each sow were done for each body posture change. Sows had, e.g. for piglet location, imbalanced repetitions from 67 to 266 observations. These random repetitions were considered as g-side random effect in the model (variance-components), without arbitrarily increasing the samples size. Examination day was taken into account as r-side random effect with compound symmetry correlation matrix.

Due to significant interaction terms of farrowing system and examination day for the three dependent variables, pairwise comparisons between the factor levels of each factor were calculated stratified according to the levels of the other factor. For estimation of the generalized linear mixed models (GLMM) including random effects and correlated errors, the procedure GLIMMIX was used.

Descriptive statistics were used to present the results for main and sub categories of body posture changes that crushed piglets, the age of the piglets at crushing and the litter behaviour during crushing events.

## 3. Results

### 3.1. Piglet Losses

The results for piglets born alive and weaned piglets are displayed in Table 1. Overall, piglet losses were analysed for 2338 piglets born alive (FC: n = 791, LH: n = 696; GH: n = 851) in nine batches. The total piglet losses in the two unrestricted farrowing systems were higher compared to pens with farrowing crates (FC: 12.3% (n = 97); LH: 25.6% (n = 178); GH: 19.9% (n = 169), *p* < 0.05). The proportion of crushed piglets relating to total piglet losses was higher in LH and GH compared to FC (FC 36.1% (n = 35), LH: 70.8% (n = 126); GH: 70.4% (n = 119), *p* < 0.05). No differences were found for total piglet mortality and the proportion of crushed piglets relating to total piglet losses between both loose-housing pens (*p* > 0.05).

### 3.2. Piglet Crushing

One hundred and three piglet crushing events were analysed (LH: n = 44 of 14 sows, GH: n = 59 of 28 sows). In both pens, rolling behaviour was the most common cause of piglet crushing, but the proportions differed between the two tested pens. In LH, crushing due to “rolling” was by far the main cause with 68.2% (n = 30), followed by “sit-to-lie” (18.2% (n = 8)), “stand-to-lie” (11.4% (n = 5)) and other reasons (2.3% (n = 1)). In contrast to LH, causes of crushing were more diverse in GH. Only 38.2% (n = 21) of all crushing in the GH-pens (n = 55) occurred due to “rolling”, followed by “sit-to-lie” (30.9% (n = 17)), “stand-to-lie” (16.4% (n = 9)) and “others” (14.5% (n = 8)). The most common cause of death in the category “others” was a sow stepping on a piglet. Four GH-piglets were crushed in the common area after co-mingling and, therefore, not regarded for the further analysis of categories and sub-categories of body posture changes that crushed piglets.

The data on piglet age at crushing was based on 97 crushed piglets (LH: n = 41; GH: n = 56). Due to technical problems, the birth of the first piglet was not able to be analysed for six piglets that were subsequently excluded from the analysis. The results highlight that the majority of crushing events occurred during the first three days postpartum in both loose housing pens (LH: 92.7% (n = 38), GH: 83.9% (n = 47)). While the proportions of crushed piglets were on a similar level for the first three days postpartum in LH (day 1: 36.6% (n = 15); day 2: 29.3% (n = 12); day 3: 26.8% (n = 11)), a peak of crushing events was found in GH on day 1 postpartum (day 1: 55.4% (n = 31); day 2: 16.1% (n = 9); day 3: 12.5% (n = 7)). Crushing was rarely seen in the remainder of lactation in both systems (LH: day 4: 4.9% (n = 2); day 5 or older: 2.4% (n = 1); GH: day 4: 3.6% (n = 2); day 5 or older: 12.5% (n = 7)). The opening of GH-pens which allowed co-mingling of sows and their litters also had no influence on the amount of piglet crushing.

The results for the sub-categories of body posture changes that lead to crushing deaths are shown in Table 2. With regard to rolling behaviour, the sub-category that most often lead to piglet crushing was a transition from sternal recumbency to the contralateral recumbency. The additional recorded behaviour of piglets during crushing events is presented in Table 3. Both in LH and GH, crushing mainly occurred when the majority of the litter were present in the pen (i.e., not in the creep area). Whereas GH-litters were mainly scored as active immediately before a crushing event, LH-litters were more often scored as inactive. Additionally, the results of piglets’ resting behaviour immediately before crushing due to rolling showed that LH-piglets more often rested in close proximity to the sow compared to GH-piglets. 

### 3.3. Sow Behaviour 72 H Postpartum

#### 3.3.1. Mean Body Posture Change Frequency

An overview of the results for the mean body posture change frequency per hour and per sow (hereinafter “mean body posture change frequency”) is given in Figure 4. The mean body posture change frequency was relatively high on day 1 postpartum compared to the two other observation days (mean body posture change frequency and SEM on day 1: FC: 3.13 (±0.39); LH: 3.40 (±0.44); GH: 2.84 (±0.37)). The mean body posture change frequency declined in all three farrowing systems between day 1 and day 2 postpartum. (mean body posture change frequency and SEM on day 2: FC: 2.22 (±0.32); LH: 2.67 (±0.28); GH 2.28 (±0.28)). On day 3 postpartum, a slight increase was observed in all three systems (mean body posture change frequency and SEM on day 3: FC: 2.47 (±0.24); LH: 2.97 (±0.30); GH: 2.78 (±0.21)). The results of the three-way mixed model analysis of variance showed a significant three-way interaction term between farrowing system, examination day and parity. Therefore, the influence of each factor was calculated by stratifying for the two other effects. Pairwise analyses of each factor with both other factors did not reveal any coherent effect of parity in the different housing systems or on the different examination days. Furthermore, no coherent effect of the farrowing system was seen in different parities or on different examination days. The examination day showed a significant effect on body posture change frequency for FC-sows of the third parity class (parity three or higher) between day 1 and day 2 and between day 1 and day 3 (*p* < 0.05) suggesting that FC-sows of parity class three performed more body posture changes on day 1 compared to day 2 and day 3. Additionally, LH-sows of the second parity performed more body posture changes at day 1 compared to day 2. An effect of the examination day could not be determined for GH-sows (*p* > 0.05).

#### 3.3.2. Frequencies of Single Body Posture Changes

A complete overview on the frequencies of single body posture changes is given in Table 4. No differences in the frequencies of the nine single body posture changes were detectable among the three farrowing systems (*p* > 0.05). Parity showed an impact on one body posture change: The body posture change “lateral recumbency to sternal recumbency” was performed more frequently by first parity sows compared to sows of the third or higher parities (*p* < 0.05). An interaction effect between farrowing system and parity was never seen (*p* > 0.05).

#### 3.3.3. Total Time Spent in Different Body Postures

Table 5 provides an overview of the percentages that sows spent in the four different body postures on the three examination days. Differences were found solely between FC and LH for the body postures “lateral recumbency”, “sternal recumbency” and “standing” on day 1 and day 3 (*p* < 0.05). The analysed three days postpartum were characterized by general inactivity. In all three farrowing systems, sows spent more than 90% of their time in a lying position (lateral recumbency or sternal recumbency). No influence of the parity could be found for the total time that sows spent in the four body postures (*p* > 0.05).

### 3.4. Piglet Behaviour 72 H Postpartum

The results for the piglet behaviour 72 h postpartum are presented in Table 6.

#### 3.4.1. Piglet Location

The probability for the location “pen” differed significantly between LH and the reference farrowing system GH on all three examination days (*p* < 0.05), i.e., the acceptance of the creep area was higher in GH compared to LH on all examination days (Table 6). Additionally, the probability for the location “pen” did not differ for LH between days 2 and 3 and the reference day 1 (*p* > 0.05), but for GH (*p* < 0.05). While the acceptance of the creep area remained at a constant low level for LH-piglets, GH-piglets visited the creep area more often as time progressed.

#### 3.4.2. Active/Inactive-Scoring of the Litter during Body Posture Changes of the Sow

The farrowing system showed no effect on day 1 (*p* > 0.05), but on day 2 and day 3 (*p* < 0.05), i.e., while the active/inactive scoring was similar for LH- and GH-piglets on day 1, it differed on day 2 and 3. LH-litters were more often scored as inactive compared to GH-litters on day 2 and day 3 (Table 6). The probability for being “active” differed in LH between the reference day 1 and days 2 and 3, i.e., LH-piglets were more often scored as “inactive” on day 2 and day 3 compared to day 1. However, no difference was found for the probability of being “active” in GH-piglets between reference day 1 and days 2 and 3 (*p* > 0.05).

#### 3.4.3. Resting Behaviour next to the Sow Immediately before Rolling Behaviour

The farrowing system showed an effect on days 2 and 3 (*p* < 0.05), but not on day 1 (*p* > 0.05), i.e., LH-piglets showed more resting behaviour next to the sow before rolling behaviour compared to GH-piglets on days 2 and 3. Whereas the probability for the event “resting” was different between day 2 and reference day 1 (*p* < 0.05), no difference existed between day 3 and reference day 1 in LH (*p* > 0.05). The probability for the event “resting” did not differ in GH between days 2 and 3 compared to the reference day 1 (*p* > 0.05), suggesting that the resting behaviour next to the sow immediately before rolling behaviour was similar on all three examination days in GH.

## 4. Discussion

### 4.1. Piglet Losses

The piglet losses were significantly higher in the two alternative farrowing systems compared to the farrowing pens with farrowing crate. This is in accordance with prior research that found higher pre-weaning mortality in free-farrowing pens compared to pens with farrowing crate [9,10]. Nevertheless, studies exist that found no higher total piglet mortality between free-farrowing pens and pens with farrowing crates [18,19,20]. Weber et al. [18] found higher levels of piglet crushing in loose housed sows compared to sows in farrowing crates and this was also the reason for the higher piglet losses in the free farrowing pens in our study. However, comparable levels of piglet losses for a group housing system were found by Li et al. [22], but the authors noticed that piglet mortality decreased over the years. This was explained by increased experience of the staff with the group housing system. As the two free-farrowing pens had just been installed before the beginning of the experiment in our study, this factor could partly explain the very high piglet losses.

### 4.2. Sow Behaviour 72 H Postpartum

The detected mean body posture change frequency was similar among the three farrowing systems at the three observation days. This is in accordance with prior research that found no differences in the number of sow body movements per hour between sows in pens with farrowing crates and loose-housed sows at the day of farrowing and the three days postpartum [17]. The descriptive analysis revealed a decline in the frequency of body posture changes for all farrowing systems in the present study between the first two observation days. Similar results were found in the above mentioned study of Weary et al. [17] and for group housed sows by Marchant et al. [23]. The most likely explanation for the decline in the overall frequency of body posture changes on day 2 is the physically demanding birth. Pain and stress could have led to an elevated frequency of body posture changes during birth on the one hand and, on the other hand, physical exhaustion could have led to a decreased frequency after birth. A phase of birth that is characterized by inactivity after a period of pre-farrowing restlessness was described by Fraser [24]. These two phases were also seen in our study, but the onset of the phase of inactivity was variable among sows. Most of the sows in our study were still in the period of restlessness when the first piglets were born, and performed many body posture changes at the beginning of the observation period. However, the decrease in body posture change frequency postpartum could also be an expression of general inactivity after birth due to natural behaviour. FC-sows of high parities (parity number three or higher) performed significantly more body posture changes on day 1 compared to day 2 and day 3. This was not seen in the two free-farrowing pens. Sows with a high parity number could have benefited from a free-farrowing pen by being encouraged to become more active earlier after birth compared to restricted FC-sows.

Significant differences for the total time that sows spent in the four body postures were found between FC- and LH-sows. The proportions for “sternal recumbency” and “standing” were significantly higher for LH-sows compared to FC-sows on day 1 and day 3, whereas FC-sows spent significantly more time in “lateral recumbency” on these two days. Sows in sternal recumbency are often awake, attentive and able to interact, for example, with their piglets or manipulable material. In contrast, sows in lateral recumbency often behave more passively. The freedom of movement and the improved possibilities to exercise natural behaviour could be the reason for the changed lying behaviour and the higher proportions of standing in the large LH-pens (7.3 m²). The assumed physical exhaustion after birth could have masked this effect on day 2 postpartum. However, no differences between FC and the smaller GH-pens (5 m²) were found, and, therefore, no general statement can be made for loose housing systems. Interestingly, differences in the active and resting behaviour became apparent among all tested farrowing systems in the course of lactation [25]. Despite the found differences between FC and LH in the present study, it must be kept in mind that the sows in all systems lay more than 90% in the first three days postpartum. Hence, the periparturient period was characterized by inactivity, and the offered possibility for more active behaviour was hardly taken up in the two free-farrowing pens. Our findings are confirmed by recent research that reported similar levels of lying for crated sows on the second day postpartum [26]. Besides being a sign of exhaustion, inactivity after birth expressed by a reduced body posture change frequency and a high proportion of lying could be an evolutionary behaviour adaption to minimize piglet losses due to crushing.

### 4.3. Piglet Behaviour 72 H Postpartum

The results of the location choices of the piglets emphasize the different levels of creep area acceptance in the two single loose-housing pens. The acceptance of the creep area was low for both GH and LH on day 1 postpartum. It increased to a higher level in GH on days 2 and 3, whilst remaining low in LH. Jensen et al [27] showed that the sow and their piglets permanently stayed together in the nest-site in a semi-natural environment for the first days postpartum, i.e., it is part of piglets’ natural behaviour to stay close to their mother sow during the first days after birth. Therefore, the attempt of modern pig husbandry to attract piglets to a separated creep area away from the sow does not correspond to piglets’ natural behaviour in this period.

Regarding this and our results for GH-pens, former research reported an initial poor acceptance of the creep area and a subsequent increase on day 3 postpartum [28,29]. Furthermore, the proportion of piglets that stayed close to the sow was initially high and subsequently decreased in the following days in earlier studies [28,29,30]. This particular behaviour seems to be deeply rooted and difficult to influence. Previous research failed at the attempt to influence this piglet behaviour by increasing the comfort and attractiveness of the creep area [31]. Moreover, former research revealed that piglets of loose-housed sows spent even more time outside the creep area postpartum compared to piglets of crated sows and the authors concluded that increased possibilities of sow-piglet interaction could be a reason for this [32].

In contrast to GH-pens, the acceptance of the creep area in LH-pens stayed at a low level throughout the three days postpartum. One possible explanation for this finding could be the higher space allowance of the LH-pens along with the fact that the creep area was located in an area that was only accessible for the piglets. This lead to a relatively large distance between the sow and the creep area.The LH-sows were not able to lie close to the creep area, which could have facilitated the acceptance of the creep area. The greater the distance between sow and creep area, the less motivated the piglets are to leave the sow and their littermates to enter the creep area after a nursing bout. The attractiveness of littermates is possibly higher than that of a distant creep area. This assumption is supported by former research that showed that piglets prefer to lie near to an anesthetized littermate in a cold part of a pen instead of lying alone in a heated creep area [33].

The different flooring in LH- and GH-pens could be a further reason for the lower acceptance of the creep area in LH-pens compared to GH-pens. The floor in GH-pens was divided into a part with fully-slatted concrete floor and a part with a fully-slatted cast-iron floor. These materials do have a high thermal conductivity compared to the plastic floor in LH-pens. This could have caused thermal discomfort for GH-piglets and could have subsequently forced the piglets to leave the pen and go to the heated creep area. In contrast, LH-piglets remained in the pen due to the more comfortable plastic flooring.

The importance of an accepted creep area for piglet survival is demonstrated by our results for piglet behaviour immediately before piglet crushing. In fact, most crushing events took place in both loose housing pens, when more than half of the litter was in the pen. Due to the increased presence of LH-piglets in the pen, piglets expressed more resting behaviour next to the sow compared to GH-pens on days 2 and 3 postpartum. The results of litter behaviour prior to crushing also suggest that LH-piglets were more often inactive in the pen compared to GH-litters. Resting behaviour in the pen is risky for piglets, because they are not able to react quickly enough to a hazardous body posture change of the sow and thus a crushing death becomes more likely.

Furthermore, the gathered data for piglets’ resting behaviour next to the sow prior to rolling events suggest that LH-piglets more often rested in close proximity to the sow immediately before the sow performed rolling behaviour compared to GH-piglets on day 2 and day 3 postpartum. The recorded piglet behaviour directly before crushing events also showed that LH-piglets rested next to the sow prior to crushing in the majority of crushing events due to rolling. Therefore, it is likely that the poor acceptance of the creep area in LH-pens lead to increased levels of resting behaviour in the pens in close proximity to the sow and subsequently to the elevated levels of crushing due to rolling.

### 4.4. Piglet Crushing

The type of body posture change that accounted for the majority of piglet crushing events in both tested loose housing pens was rolling behaviour, followed by a transition from “sitting to lying” and the movement “standing to lying”. Rolling behaviour as the main cause for piglet crushing in loose-housing pens was also reported elsewhere [17,34]. Nevertheless, in most studies dealing with this topic, a transition from “standing to lying” was the main reason for crushing of piglets [23,35,36]. However, interestingly, this postural change was only responsible for a minority of crushing deaths in our study. A large influence of individual sows on the detected posture changes that crushed piglets can be assumed: the total number of crushed piglets was different between individual sows. It is conceivable that particular body posture changes are more risky in some sows than in others and subsequently lead to more crushing deaths. Large differences in the extent of piglet crushing were also observed by Jarvis et al. [37] and can be explained with individual behavioural differences between sows regarding maternal behaviour [38] and responsiveness to situations that are dangerous for piglets [11]. Furthermore, individual differences may have had a greater influence on our results for LH than for GH, because the number of observed sows was smaller in LH compared to GH. Since the two loose-housing systems were installed just before the experiment started, all multiparous sows farrowed in pens with farrowing crate in their previous gestations in our study. However, recent research showed that loose-housed sows lost fewer piglets due to crushing, if they had already experienced the same system in the previous lactation [39]. Therefore, it is possible that our results for piglet crushing would be different, if the sows had already experienced the two free-farrowing pens before the start of the experiment.

Interestingly, different crushing patterns were found for LH- and GH-pens. The elevated levels for crushing due to rolling behaviour in LH-pens compared to GH-pens can be explained by the already discussed piglet behaviour. The most dangerous rolling behaviour in both free-farrowing pens was a transition from sternal recumbency to lateral recumbency with a simultaneous side swap. The authors of the study often observed that piglets preferred to lie near the belly side/udder side of a sow that lies in asymmetric sternal recumbency, instead of lying near the dorsal side. Therefore, it is more likely that a crushing event occurs if a sow changes from an asymmetric sternal recumbency to a lateral recumbency in the direction of her belly side (i.e.,swapping side) compared to a change from sternal to lateral recumbency in the direction of the dorsal side (i.e., without swapping side). The crushing category “others” mainly contained crushing events caused by accidental trampling on a piglet, and this was especially observed in GH-pens. Additionally, high levels of crushing due to a transition from sitting to lying were observed in GH-pens. A possible explanation for these findings could be the relatively small size of the GH-pens (approximately 5 m²) compared to the LH-pens (approximately 7m²). In GH-pens, piglets that remained in the pen were automatically in the activity area of the sow. They were not easily able to increase the distance to the dangerous zone around the sow, whereas LH-pens were larger and contained an area that was only accessible for piglets. Subsequently, the probability was higher in GH-pens that piglets were in immediate vicinity of the sow.Therefore, it was more likely that accidental trampling on a piglet happened or that a sow crushed a piglet during transition from a sitting to a lying position.

The majority of crushing events were recorded in LH- and GH-pens in the first three days postpartum. This is in accordance with Marchant et al. [15] who recorded the majority of piglet crushing in the first four days postpartum. A peak for piglet crushing existed in GH-pens on day 1 postpartum, whereas the percentages of crushed piglets were similar in LH-pens for the first three days postpartum. The peak of crushing on the first day postpartum that was found for GH-pens was also described in previous research [15,17,23] and can be explained with the already discussed piglet behaviour concerning the acceptance of the creep area on day one. The acceptance of the protective creep area was low on day 1, and subsequently more piglets were in the hazardous pen during body posture changes of the sow. The acceptance increased in GH-pens on day 2 and day 3 and could be the reason for the reduced number of crushing deaths on these two days. The fewer piglets are present in the pen, the less likely it is that a piglet will be crushed during a body posture change of the sow. The acceptance of the creep area was not only low for LH-litters on day 1, but remained low on the two subsequent days. These findings explain why the number of crushing deaths was at a similar level in LH-pens for the first three days postpartum.

Our study confirmed that the first three days after birth are the most dangerous days for suckling piglets in a loose-housing pen with regard to piglet crushing. Additionally, the conclusion can be drawn that the risk of being crushed is at a low level in the remaining lactation period. Even though piglet losses were higher in our both tested single loose-housing pens compared to FC, our results indicate that confining the sow for the entire lactation period may not be necessary. A temporary confinement after birth could be a practical compromise for sows’ and piglets’ welfare. This is supported by former research showing that a four day period of fixation after birth was sufficient to reduce piglet mortality compared to loose-housing pens [40]. The above discussed results for the sow behaviour 72 h postpartum showed that the postpartum period was characterized by inactivity and that the influence of the housing environment on sow behaviour was low. These findings suggest that the farrowing crate did not hinder the sow in the expression of her natural active and resting behaviour in the 72 h postpartum This could be an argument justifying the sows’ fixation for a couple of days after birth. Nonetheless, it must be kept in mind that this statement is only valid for the active and resting behaviour in the postpartum period. It is known that sows have an increased urge for movement in the prepartum period [27]. Hales et al. [41] only detected a difference in total piglet mortality between loose housed sows and temporary fixated sows when the sow had already been confined before birth, i.e., in the period in which sows are highly motivated to move. Additionally, it is known that the farrowing crate restricts more behaviours than just locomotion (e.g., defecation behaviour, thermoregulation, nest-building [3]). Therefore, a temporary confinement for a couple of days after birth should only be a transitional solution until improvements in breeding and the design of free-farrowing pens allow farrowing without any fixation.

## 5. Conclusions

In conclusion, the 72 h period after birth was characterized by inactivity of the sows which seemed to be independent of the tested farrowing environment. Differences in the active and resting behaviour of sows were rarely detected among the three systems in this period. Piglet behaviour differed considerably between both tested free-farrowing pens and could explain the different crushing patterns found between LH and GH. Most piglet crushing occurred during the first three days after birth, and a temporary fixation of the sow during and after birth could help to reduce piglet losses due to crushing. With the detected behaviour patterns both in sows and piglets, this study highlights the importance of a proper pen design to particularly influence piglet behaviour after birth and subsequently reduce piglet crushing.

## Figures and Tables

**Figure 1 animals-09-00538-f001:**
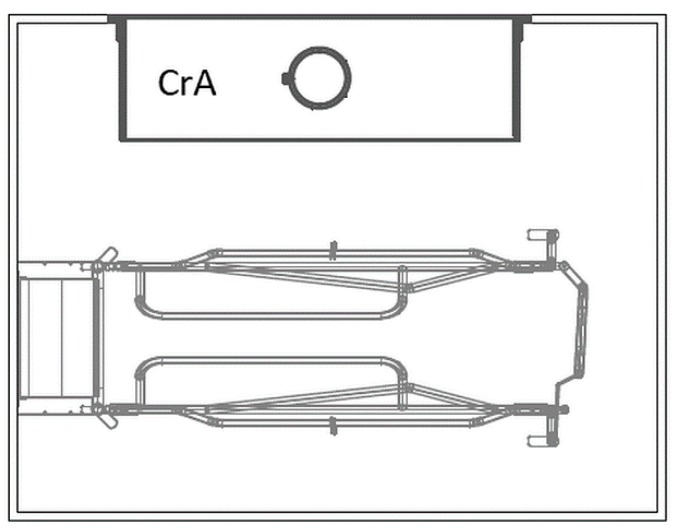
Pen with farrowing crate (FC). CrA = creep area. © Big Dutchman International GmbH, Vechta, Germany.

**Figure 2 animals-09-00538-f002:**
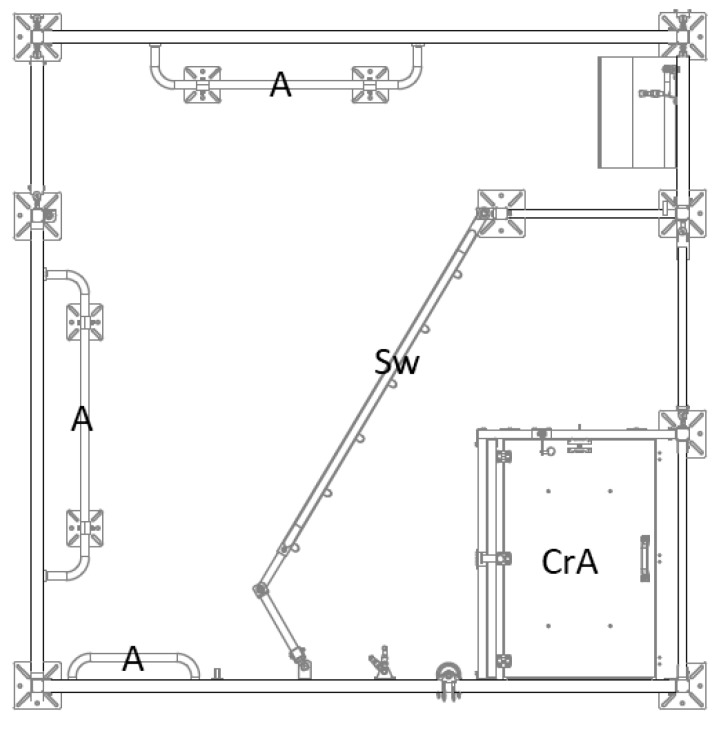
Single loose housing pen (LH). CrA = creep area, Sw = swing gate, A = anti-crushing bars. © Big Dutchman International GmbH, Vechta, Germany.

**Figure 3 animals-09-00538-f003:**
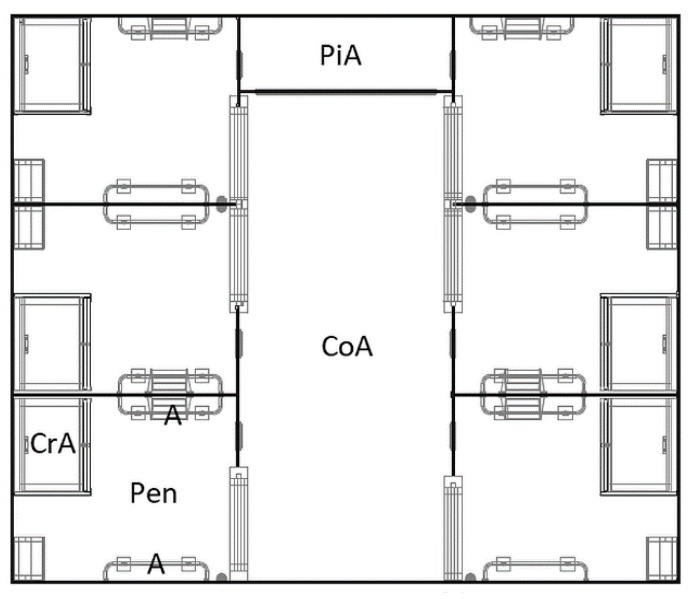
Group-housing system for lactating sows (GH). CrA = creep area, A = anti-crushing bar, CoA = common area, PiA = piglet area. © Big Dutchman International GmbH, Vechta, Germany.

**Figure 4 animals-09-00538-f004:**
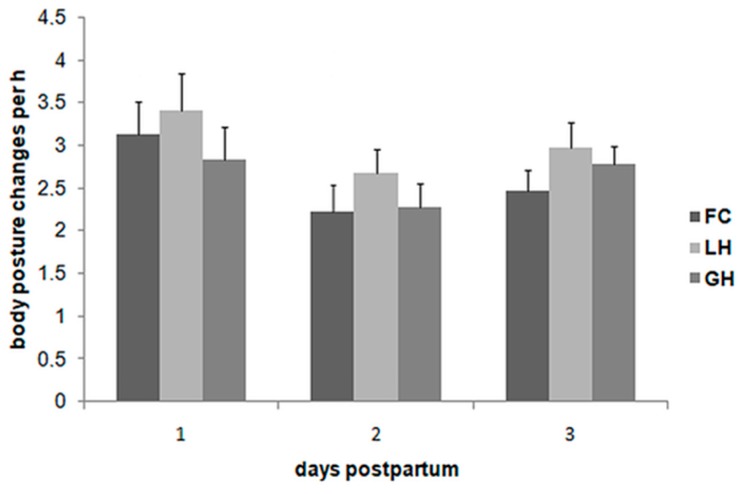
Mean body posture change frequencies (per sow and per hour) in the different farrowing systems (conventional pen with farrowing crate (FC), single loose housing pen (LH), single loose housing pen of the group housing system (GH)) on the three days postpartum.

**Table 1 animals-09-00538-t001:** Mean values and standard error of the mean for piglets born alive per litter, weaned piglets per litter and piglet losses per litter. FC = pens with farrowing crate, LH = single loose housing pens, GH = group housing system. Different superscripted letters (a, b) mark significant differences between the systems.

Farrowing System	Total Number of Litters	Total Number of Piglets	Birth Weight (kg)	Piglets Born Alive	Weaned Piglets	Piglet Losses	Piglet Losses (%)	Proportions of Crushed Piglets (%)
FC	53	791	1.39 (±0.01)	14.9 (±0.60)	13.1 (±0.54)	1.8 (±0.21) ^a^	12.3 ^a^	36.1 ^a^
LH	49	696	1.45 (±0.01)	14.2 (±0.59)	10.6 (±0.48)	3.6 (±0.47) ^b^	25.6 ^b^	70.8 ^b^
GH	54	851	1.42 (±0.01)	15.8 (±0.52)	12.6 (±0.38)	3.1 (±0.33) ^b^	19.9 ^b^	70.4 ^b^

**Table 2 animals-09-00538-t002:** Types and numbers of sows’ body posture changes that crushed piglets in single loose housing pens (LH; n= 44) and pens of the group housing system (GH; n = 55). Flopping means an uncontrolled fall down of the sow. Four GH-piglets were crushed in the common area after co-mingling and are not regarded in this table.

Main-Category	Sub-Category	LH	GH
Stand-to-lie	Lay down in lateral recumbency	0	0
	Lay down in sternal recumbency	2	4
	“flopping” in lateral recumbency	3	5
Sit-to-lie	Lay down in lateral recumbency	2	2
	Lay down in sternal recumbency	4	5
	“flopping” in lateral recumbency	2	10
Rolling	Lateral recumbency to sternal recumbency	4	3
	Sternal recumbency to lateral recumbency (central position)	5	4
	Sternal recumbency to lateral recumbency (same side)	5	2
	Sternal recumbency to lateral recumbency (swap side)	15	8
	Lateral recumbency to lateral recumbency (swap side)	0	2
	Sternal recumbency to sternal recumbency	1	2
Others		1	8

**Table 3 animals-09-00538-t003:** Number of recorded litter behaviour during crushing events in single loose housing pens (LH) and pens of the group housing system (GH). Location (LH: n = 42 of 13 litters, GH: n = 48 of 25 litters), active/inactive (LH: n = 42 of 13 litters, GH: n = 51of 25 litters), resting behaviour next to the sow during rolling (LH: n = 27 of nine litters, GH: n = 21 of 15 litters).

RecordedLitter Behaviour	Definition of Behaviours	LH	GH
Location	≥50% of the litter in the creep area	2	6
	>50% of the litter in the pen	40	42
Active/inactive	>50% of the piglets in the pen are active	19	45
	≥50% of piglets in the pen are inactive	23	6
Resting behaviour next to the sow during rolling	No piglet rests next to the sow’s body immediate to the rolling event	4	15
	At least one piglet rests next to the sow’s body immediate to the rolling event	23	6

**Table 4 animals-09-00538-t004:** Mean frequencies (per 24 h) and standard error of the mean (SEM) of body posture changes in the three tested farrowing systems: pens with farrowing crate (FC), single loose housing pen (LH), single loose housing pen of the group housing system (GH). No significant differences were found among systems (*p* > 0.05).

Body Posture Change	FC	LH	GH
Stand to sternal recumbency	6.29 (±0.82)	5.99 (±0.58)	4.07 (±0.62)
Stand to lateral recumbency	0.91 (±0.29)	2.73 (±0.68)	2.53 (±0.74)
Sit to sternal recumbency	11.12 (±1.25)	9.63 (±2.28)	7.73 (±1.27)
Sit to lateral recumbency	0.87 (±0.28)	1.83 (±0.47)	1.66 (±0.58)
Lie to stand	0.71 (±0.22)	1.51 (±0.90)	1.26 (±0.40)
Lie to sit	17.67 (±1.47)	17.90 (±2.64)	14.28 (±1.76)
Sit to stand	6.11 (±0.87)	7.07 (±0.73)	5.40 (±0.85)
Sternal recumbency to lateral recumbency	11.71 (±1.42)	14.02 (±2.09)	13.94 (±1.61)
Lateral recumbency to sternal recumbency	6.91 (±1.96)	11.81 (±2.15)	13.85 (±2.33)

**Table 5 animals-09-00538-t005:** Mean percentage of sows’ body postures on different examination days. Farrowing system: FC = pen with farrowing crate; LH = single loose housing pen; GH = single loose housing pen of the group housing system. Body posture: LR = lateral recumbency; SR = sternal recumbency; TL = total lying; Si = sitting; St = standing. Significant differences between farrowing systems within one examination day are marked by different superscripted letters (a, b).

		Body Posture				
Examination Day	Farrowing System	LR (%)	SR (%)	TL (%)	Si (%)	St (%)
Day 1	FC	89.7 ^a^	6.7 ^a^	96.4	1.4 ^a^	2.2 ^a^
	LH	81.8 ^b^	11.6 ^b^	93.4	1.0 ^a^	5.5 ^b^
	GH	85.2 ^a,b^	9.8 ^a,b^	95.0	1.2 ^a^	3.8 ^a,b^
Day 2	FC	82.3 ^a^	12.6 ^a^	94.9	1.0 ^a^	4.1 ^a^
	LH	76.9 ^a^	15.7 ^a^	92.6	1.3 ^a^	6.1 ^a^
	GH	80.6 ^a^	14.1 ^a^	94.7	1.0 ^a^	4.3 ^a^
Day 3	FC	81.1 ^a^	12.3 ^a^	93.4	1.4 ^a^	5.1 ^a^
	LH	73.1 ^b^	18.5 ^b^	91.6	1.5 ^a^	7.0 ^b^
	GH	76.6 ^a,b^	16.5 ^a^	93.1	1.0 ^a^	6.0 ^a,b^

**Table 6 animals-09-00538-t006:** Percentages of recorded litter behaviour (location, active/inactive, resting behaviour next to the sow immediately before rolling) on different examination days. Farrowing systems: FC = pen with farrowing crate; LH = single loose housing pen; GH = single loose housing pen of the group housing system.

Recorded Litter Behaviour		day 1		day 2		day 3	
		LH	GH	LH	GH	LH	GH
Location	Pen	98.8% (n = 417)	90.9% (n = 280)	98.4% (n = 485)	52.9% (n = 262)	94.0% (n = 594)	38.6% (n = 210)
	Creep area	1.2% (n = 5)	9.1% (n = 28)	1.6% (n = 8)	47.1% (n = 233)	6.0% (n = 38)	61.4% (n = 334)
Active/inactive	Active	89.7% (n = 376)	89.9% (n = 276)	74.9% (n = 364)	91.5% (n = 314)	71.1% (n = 424)	93.2% (n = 273)
	Inactive	10.3% (n = 43)	10.1% (n = 31)	25.1% (n = 122)	8.5% (n = 29)	28.9% (n = 172)	6.8% (n = 20)
Resting behaviour	Resting	21.1% (n = 32)	13.8% (n = 17)	38.7% (n = 67)	8.8% (n = 17)	24.1% (n = 57)	7.6% (n = 20)
	No resting	78.9% (n = 120)	86.2% (n = 106)	61.3% (n = 106)	91.2% (n = 177)	75.9% (n = 180)	92.4% (n = 243)

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
