# Peer review of "The Effect of Sows’ and Piglets’ Behaviour on Piglet Crushing Patterns in Two Different Farrowing Pen Systems"

_animals, 2019, doi:10.3390/ani9080538_

Round 1

Reviewer 1 Report

The paper has obviously taken advantage of a new housing system that has become available. You have analysed the effect of this new housing system on the mortality rate of piglets by crushing. At the moment, the paper is incredibly complicated, more so than it needs to be. It is quite hard to follow the exact numbers used and how the design was actually carried out. I am not sure exactly what makes GH and LH different given that GH were confined separately for up to five days, and you only watched them for 3 days.

I would suggest that you have a go at re-writing the paper making it much more concise and to the point that you are trying to make. You may also want to ask a native English speaker to check your grammar, sentence and paragraph structure. Try to make your sentences no more than two lines in length, and no paragraph should be just one sentence long. You also need to make sure your tables are clear and that they have all been referred to. An ethogram of your behaviours would be beneficial. More detail is required in your statistical reporting and not just the p-values. 

At present, I do not see the novelty in your work and how it is adding something new to the literature. I am also concerned that piglets were teeth clipped and tail docked at 4 days old. According to the COUNCIL DIRECTIVE 2008/120/EC, made on 18 December 2008 laying down minimum standards for the protection of pigs, Annex to Directive 98/58/EC, the following requirements apply: “Neither tail-docking nor reduction of corner teeth must be carried out routinely but only where there is evidence that injuries to sows’ teats or to other pigs’ ears or tails have occurred. Before carrying out these procedures, other measures shall be taken to prevent tail-biting and other vices, taking into account environment and stocking densities. For this reason inadequate environmental conditions or management systems must be changed.” Can you provide evidence that this was needed at 4 days old in these piglets and that you had made significant changes to the accommodation of the piglets that was trying to prevent this from happening?

Reviewer 2 Report

The article as a whole presents important information regarding the impact of various housing systems on piglet crushing in sows.  The authors do a good job of describing the specific types of movements involved in piglet crushing and highlighting that different behaviors are involved in piglet crushing in the different housing systems.  There are a few study design issues that must be addressed, and omission of parity and sow experience may be influencing their results.  The first sentence of the simple summary is clunky and awkward.  I understand what the authors are trying to say, but “diversity of natural behavior patterns” does not accurately represent the situation.  Suggest revision. The abstract does not clearly define what LH and GH are.  Therefore, it is difficult to interpret the remainder of the abstract.  What makes them different? Lines 51-53.  Suggest breaking this into multiple sentences.  Will improve readability.  Further, the authors should acknowledge that even though consumers SAY they are willing to pay more for improved welfare, they rarely do. There must be acknowledgement and information provided in the text and in the statistical analysis of parity of the sows in the study.  Parity plays a huge role in piglet crushing behavior, yet is overlooked in the analysis.  What proportion of piglet crushing sows were of what parity?  Were the parities evenly distributed across treatments?  If a first parity gilt was housed with experienced sows in the LH, then did they perform differently than gilts that were not exposed to experienced conspecifics?’ Table 4: there are no statistical differences among any of the treatments for any of the variables.  Perhaps add a p-value column instead of putting all lowercase letters? Figure 2: the way it reads, day 1 is different from day 2 for FC sows only.  I would expect day 1 to also differ from day 3.  Is there a missing piece of this figure?  This differs from the text.  Please resolve.  Also, is it body posture changes per/h/sow, or for all sows combined?  Just want to have clarity on the units. 446-451: these conditions regarding sow experience and uneven allotment of sows based upon previous experience and parity presents a study design flaw that may be confounding the results.  This must be resolved.

Reviewer 3 Report

The paper deals with an important topic and provides new information with regards to farrowing systems which may be increasingly  used in the future. Some of the results had already been presented before, but the paper provides deeper insights in some specific aspects (specific body changes producing more crushing of piglets, for example). In general, it is a clear paper, and discussion is adjusted to the results.

There are some aspects which remain unclear or could be improved.

Specific comments:

-          Introduction

Line 55-60. Really long sentence.

-          Material and Methods

Line 128-130. Could authors provide a bit more of information/precision on the enrichment materials provided to sows/piglets (number of ropes, jute sacks..). Were they balanced across the different systems? Were they available for sows before parturition?

Line 135. Sows in the GH-system were fed on demand means “ad libitum” or the amount of feed they could demand was restricted to the same level as for the other two systems fed twice a day?  (for example, they received 0.5kg more from day 17).  The authors have not discussed differences on feeding behaviour, it was not their objective, but do they think that being able to feed on demand could have a positive effect for GH sows as compared to LH or FC sows?

Figure 1a-1c. If some of the elements like “protection bars”, or the outdoor creep area in the GH system were labelled, it would make figures easier to understand.

Line 164-167. Piglets were weighed when they were born. Could the authors provide a mean of the newborn piglets for the different farrowing systems? The genetics used was in the limit to being considered hyperprolific (15 piglets born alive) and weigh of the piglet is a critical issue with regards to its survival.

Line 175-178. The total number of born alive was already lower in the LH system, although according to Table 1, not significantly, but do the authors think that the system could have also an influence on this parameter (if more sows had been tested)?

Line 178. Could the authors provide more detail on how a piglet was identified as dead due to crushing, to differentiate from stillborns for example? What signs used the trained staff.

Line 180-182. 24 sows were video recorded in the LH system to evaluate piglet crushing in more detail and 46 for the GH system. Was this because of a logistical question? The authors discuss further on that individual differences are of great relevance in the patterns/body changes that produce piglet crushing (line 443-446, and later on). Do the authors think that by having a lower number of individual sows (i.e. higher weigh of individual differences) in the LH system as compared to the GH could have any influence on the results?

Lines 213-223. More detail on the statistical analysis should be provided. For example: “considering the potential influencing factors”. These factors should be detailed. Also details on what was considered as the experimental unit, if sow was included as “random effect”, detail also the experimental unit. A model of repeated measurements were considered when analysing parameters over time in the same individuals (such as body posture changes at 1,2 and 3 days post partum)?

-          Results

             Table 1. “piglet losses” 1.8, 3.6, 3.1 refer to mean piglet losses/litter?

       Table 2. It is a bit difficult to follow/understand the results. In the text, it is stated that 103 crushing events were evaluated (LH=44 and GH=59), but in Table 2, 43 and 47 (total 90 sows/crushing events) respectively are included. The data for age of piglet crushing is based on 97 piglets, but the text does not reflect that not all the crushing events (103) could be evaluated for body changes. The difference of 13 events is due to the “other reasons” (9 according to the text)?

Table 2. Sternal recumbency to sternal recumbency, means a total loop?

Figure 2. If there were no differences within the same day, is there a need to label with letters? Maybe only mention  “No differences within the same examination day between systems”, instead of “significant differences…)(line 287)

Table 4. The title mentions “frequencies per 24h” in the text (lines 196-198) that these frequencies were obtained from observations of the first 72hours. Could the authors clarify a bit more how these frequencies were calculated?

Figure 3. The ethogram of behavioural observations seems to have collected only “body changes and postures”. Feeding, drinking…were not recorded?

-          Discussion

Line 360-366. Do the authors think that the higher number of body changes the first day post partum compared to the second day could still be associated with pain/distress caused by parturition? To what do the authors associated the fact that this decrease was only significant for FC sows?

Line 449-452. The authors seem to indicate that the fact that the sows had not experienced the system before is one of the reasons why they performed more rolling behaviour compared to other studies? How is this related with the previous explanation (line 443-446) about individual differences? It is a bit difficult to follow the rationale in this part of the discussion.

Round 2

Reviewer 2 Report

Thank you for the much improved manuscript.  I am excited for the contribution to our knowledge base.

I appreciate the authors’ statement: “There is an urgent need to find a practical compromise between sows and piglets’ welfare”.  This is a conversation that is had less often than needed, compared to the evaluation of conflicts among the different components of welfare.  While the manuscript has undergone substantial improvement, further revisions are needed.

Line 24: do the authors mean 5 m2/sow, cast iron…?

Line 25: can delete “only” before LH and GH to save on word count.

Line 102: by “compartment” do you mean “room”?

Statistical analysis: I would have anticipated that the authors would evaluate the interaction between housing system, parity, and time.  As the statistics reads, they evaluated differences either among the housing systems or among the different parity classes within each day, but did not evaluate any further.  What were the repeated factors in the model?  If the same data is collected over time, then this is a repeated measures study and needs to be analyzed as such.

Table 2: what do you include as “others”?  There’s a big difference between the two housing systems, so any insight here would be helpful.

Table 3: the layout/formatting is odd.  Can you left-justify the text?  There are weird things happening with the spacing and bolding in this table.  Perhaps a bit more TLC to all the tables to ensure clarity of message.

Figure 4: why mention the other asterisk information when only one type is provided.  I’m also a bit confused.  Is the asterisk indicating that there is a difference between FC on day 1 and day 2?  The authors might want to consider using a lettering system for clarity because they are presenting data that could have had an interaction.  Further, data collected across time is typically represented as a line graph, not a vertical column.

Table 4: the lowercase letters are not needed.  If there are no significant differences among the three housing systems, nor are there differences among the multiple behaviors presented, then these are unnecessary.  If there are no differences among the treatments, then this table might be better used to demonstrate the differences among parities, as is described in lines 306-310.

Table 5: What is “TL” and why do none of the numbers in this column have a superscript?  Do the different numbers represent differences among postures or differences among housing systems? Differences among housing systems within day or across the entire 3 days?  This needs to be in the figure title.  In general, this table needs more TLC as there are inconsistencies in formatting and several important factors remain unclear.

In general, you do not need to say “significantly”.  If there are differences, then you support that with a p-value.  Also, when you are evaluating 2 things, then you use the word “between”, if you are evaluating 3 or more things, then you use the word “among”.  Please check the entire manuscript for this.

The data presented in section 3.4 would be easier to read/interpret if it was presented as a table rather than text.  Sometimes it is hard to see the forest for the trees.

Line 389: delete “very”

Line 445: I believe the authors mean “contrast”, not “contrary”

Supplementary tables: In general, they need more TLC in formatting.  As is, they are difficult to interpret and do not provide much information to the reader.

Reviewer 3 Report

Dear Authors

The authors have addressed all comments and the paper is now acceptable for publication.

Author Response

3.1) The authors have addressed all comments and the paper is now acceptable for publication.

Thank you for your very helpful comments to our manuscript.